# Methionine γ-Lyase-Daidzein in Combination with S-Propyl-L-cysteine Sulfoxide as a Targeted Prodrug Enzyme System for Malignant Solid Tumor Xenografts

**DOI:** 10.3390/ijms231912048

**Published:** 2022-10-10

**Authors:** Louay Abo Qoura, Elena Morozova, Vitalia Kulikova, Saida Karshieva, Darina Sokolova, Vasiliy Koval, Svetlana Revtovich, Tatyana Demidkina, Vadim S. Pokrovsky

**Affiliations:** 1Department of Biochemistry, RUDN University, 117198 Moscow, Russia; 2Engelhardt Institute of Molecular Biology of the Russian Academy of Sciences, 119991 Moscow, Russia; 3Laboratory of Combined Treatment, N.N. Blokhin National Medical Research Center of Oncology of Ministry of Health of Russian Federation, 115478 Moscow, Russia; 4Department of Biotechnology, Sirius University of Science and Technology, 354340 Sochi, Russia

**Keywords:** methionine γ-lyase, daidzein, directed enzyme prodrug therapy, thiosulfinates, anticancer activity

## Abstract

The purpose of this study was to determine the anticancer effect of dipropyl thiosulfinate produced in situ by the pharmacological pair: (1) conjugated with daidzein C115H methionine γ-lyase (EC 4.4.1.11, C115H MGL-Dz) and (2) the substrate, S-propyl-L-cysteine sulfoxide (propiin) against various solid tumor types in vitro and in vivo. The MTT test was used to calculate IC_50_ values for HT29, COLO205 and HCT116 (colon cancer); Panc1 and MIA-PaCa2 (pancreatic cancer); and 22Rv1, DU-145 and PC3 (prostate cancer). The most promising effect for colon cancer cells in vitro was observed in HT29 (IC_50_ = 6.9 µM). The IC_50_ values for MIA-PaCa2 and Panc1 were 3.4 and 3.8 µM, respectively. Among prostate cancer cells, 22Rv1 was the most sensitive (IC_50_ = 5.4 µM). In vivo antitumor activity of the pharmacological pair was studied in HT29, SW620, Panc1, MIA-PaCa2 and 22Rv1 subcutaneous xenografts in BALB/c nude mice. The application of C115H MGL-Dz /propiin demonstrated a significant reduction in the tumor volume of Panc1 (TGI 67%; *p* = 0.004), MIA-PaCa2 (TGI 50%; *p* = 0.011), HT29 (TGI 51%; *p* = 0.04) and 22Rv1 (TGI 70%; *p* = 0.043) xenografts. The results suggest that the combination of C115H MGL-Dz/propiin is able to suppress tumor growth in vitro and in vivo and the use of this pharmacological pair can be considered as a new strategy for the treatment of solid tumors.

## 1. Introduction

Targeted therapy, which utilizes enzymes in cancer cells directly without damaging healthy cells, is becoming a viable alternative to conventional chemotherapy [1]. Enzymes can be used as anticancer agents or as carriers of cytotoxic drugs by directly targeting cancer cells [2]. Cytotoxic therapy is associated with severe adverse effects and limited efficacy due to a lack of targeted accumulation in cancer cells [3], and the effective cytotoxic doses causing severe damage in normal cells. The use of targeted drugs has the benefit of overcoming some of these disadvantages [4]. Prodrug activation enzymes could be considered as another approach to clear this obstacle. First, drug-activating enzymes are delivered to tumors or can be expressed in cancer cells. In the next step, harmless prodrugs, which are substrates of the aforementioned enzymes, can be administered to be converted into toxic compounds directly in tumors [5]. Daidzein is a naturally occurring isoflavone compound derived primarily from soybeans that resembles natural estrogens in structure and competes with them for estrogen receptor binding (ERs) [6]. Daidzein can bind to both ER-α and ER-β and has a dose-dependent effect on the transcription of estrogen-responsive target genes [7]. In addition, daidzein can interact with membrane G protein-coupled estrogen receptor 1 (GPER) [8]. To date, many tumors are characterized by high level of expression of these receptors, and can therefore be suitable as a targets for direct drug delivery [6,8]. It was shown that daidzein conjugates with antitumor drug daunomycin, which together bind directly to the tumor and retain their cytotoxic properties [9]. The usage of targeted pro-drug enzyme system daidzein-allianase conjugates/alliin in directed enzyme prodrug therapy (DEPT) of ovarian carcinoma was demonstrated [10]. Daidzein–alliinase conjugates have been shown to produce cytotoxic allicin from alliin in situ and selectively inhibit the growth of ovarian cancers without causing visible toxic side effects [10]. Earlier, we have demonstrated that pyridoxal 5′-phosphate dependent methionine γ-lyase (EC MGL; EC 4.4.1.11) catalyzes the breakdown of S-substituted L-cysteine sulfoxides with the formation of thiosulfinates possessing cytotoxic activity [11]. The combination of the enzyme with these substrates was defined as a “pharmacological pair”. A mutant form of the enzyme from *Clostridium novyi* with the replacement of Cys115 for His (C115H MGL) exhibited higher activity in the β-elimination reaction of S-substituted L-cysteine sulfoxides compared to the native enzyme [12] and was used in this study as a component of the pharmacological pair. Propiin (S-propyl-L-cysteine sulfoxide) was used as the second component of the pair. Its breakdown by C115H MGL produces dipropyl thiosulfinate (Figure 1), which possesses cytotoxic properties in vitro [11]. In this work, the cytotoxic effect of dipropyl thiosulfinate derived by the pharmacological pair C115H MGL-Dz/propiin on various cancer cells was demonstrated in vitro. The antitumor effect of the pharmacological pair was evaluated on colon cancer (SW620 and HT29), pancreatic cancer (MIA-PaCa2 and Panc1) and prostate cancer (22Rv1) xenografts in BALB/c nude mice models.

## 2. Results

### 2.1. Steady-State Kinetic Parameters of the β-Elimination Reaction Catalyzed by the Enzyme and Characterization of the Conjugated C115H MGL-Dz

Steady-state kinetic parameters of the γ- and β-elimination reactions catalyzed by *C. novyi* C115H MGL were previously obtained [11]. In the present study, the steady-state kinetic parameters of the β-elimination reaction of S-(allyl/alkyl)-L-cysteine sulfoxides and S-methyl-L-cysteine were determined for C115H MGL-Dz (Table 1). Three daidzein moieties were coupled to C115H MGL in the C115H MGL-Dz conjugate. The catalytic efficiency of C115H MGL-Dz in the β-elimination reaction of S-(allyl/alkyl)-L-cysteine sulfoxides proved to be almost the same as that for the non-conjugated enzyme.

### 2.2. C115H MGL-Dz Cytotoxicity on Different Cancer Cells Viability in the Presence of Propiin In Vitro

The cytotoxicity of C115H MGL-Dz in the presence of propiin was evaluated on different cancer cells. After incubation with cancer cells, the conjugate was washed to remove the non-bound conjugate, and 1 mg/mL of propiin was added, as described in “Section 4”. In the presence of propiin, the C115H MGL-Dz had a cytotoxic effect on HT29, Panc1, MIA-PaCa2 and 22Rv1 cell lines. COLO205 or SW620 were not sensitive to C115H MGL-Dz in the presence of propiin (Table 2). C115H MGL-Dz alone was not toxic against all cell lines.

### 2.3. Cytotoxicity Effect of Dipropyl Thiosulfinate on Various Cancer Cells Viability

The effect of dipropyl thiosulfinate on the tumor cells viability in comparison with C115H MGL + PBS as a control were measured on a panel of human cancer cell lines: HT29, COLO205 and HCT116 (colon cancer cell lines); Panc1 and MIA-PaCa2 (pancreatic adenocarcinoma cell lines); and 22Rv1, DU-145 and PC3 (prostate cancer). As illustrated in Table 3, dipropyl thiosulfinate exhibits a potent antiproliferative effect on a variety of cancer cell lines. Among them, Panc1 cells demonstrated the lowest IC_50_ value (18.6 µM), indicating the highest sensitivity to dipropyl thiosulfinate. For prostate cancer cells, the most promising effect of dipropyl thiosulfinate was against PC3 (IC_50_ = 51.2 µM) compared with DU-145 (61.0 µM) or 22Rv1 (66.6 µM). Compared with HT29, HCT116 or COLO 205, the cytotoxicity of the dipropyl thiosulfinate was the most valuable against SW620 colon cancer cells (IC_50_ = 19.2 µM). Propiin alone was not toxic against any cell lines and IC_50_ values were not detected.

### 2.4. SW620 and HT29 Xenografts Showed Different Responses to the Pharmacological Pair In Situ

In female nude BALB/c mice implanted with HT29 cells, treatment with the pharmacological pair (Group 2) resulted in decreased tumor growth (Figure 1A), and tumor size at the end of the experiment (275 ± 49 mm^3^; *p* = 0.04) was significantly smaller than in the control group (Group 1) (571 ± 114 mm^3^). C115H MGL-Dz/PBS (Group 3) suppressed tumor growth by 37% (*p* = 0.11), with no significant difference between experimental and control groups. On day 10, tumor growth decreased to 44% (*p* = 0.09) in Group 2 compared with Group 1, although the effect was not significant. The data are shown in Table 4. The tumor growth after administration of the pharmacological pair was suppressed by 22% (*p* = 0.12) at day 10 in male BALB/c mice implanted with SW620 cells (Figure 1B). However, C115H MGL-Dz/PBS reduced tumor growth by 15% from the first day of treatment compared to the control when the differences were not statistically significant (*p* = 0.165). On the last day of therapy, tumor growth in Group 4 (the treatment by C115H MGL/propiin) was reduced by 24% compared to the vehicle group, and the effect was not significant (*p* = 0.155).

### 2.5. The Pharmacological Pair Inhibits Growth of Pancreatic Adenocarcinoma Panc1 and MIA-PaCa2 Xenografts

The treatment of human pancreatic adenocarcinoma xenograft Panc1 in nude BALB/c mice with C115H MGL-Dz/PBS markedly reduced the volume of the tumor (228 ± 61.2 mm^3^; TGI 49%; *p* = 0.160) compared with the vehicle (442.9 ± 65.5 mm^3^; Figure 2A). Treatment using C115H MGL/propiin did not cause a significant reduction in the tumor volume (216 ± 34 mm^3^; TGI 51%; *p* = 0.462 vs. control). On the last day of therapy, the injection of the pharmacological pair resulted in a significantly higher reduction in the tumor volume (144 ± 37 mm^3^; TGI 68%; *p* = 0.004 vs. C115H MGL-Dz/PBS or C115H MGL with propiin). These results suggest that inhibition by the pharmacological pair may enhance the suppression of Panc1 xenograft growth in vivo. The initial xenograft volumes for MIA-PaCa2 cells are shown in Figure 2B. On the last day of treatment, the average tumor volume reached275 ± 72 mm^3^ in Group 1, 138 ± 44 mm^3^ in Group 2 (TGI 50%, *p* = 0.011), 197 ± 86 mm^3^ in Group 3 (TGI 28%; *p* = 0.439), and 187 ± 133 mm^3^ in Group 4 (TGI 32%, *p* = 0.606). Co-treatment with C115H MGL-Dz and propiin significantly reduced the tumor volume in mice compared to single treatment with C115H MGL-Dz or unconjugated C115H MGL with propiin in MIA-PaCa2 tumors. The inhibition rate of tumor growth in Group 2 was 50%, which was significantly higher than that of the other groups (*p* = 0.011) (Figure 2B).

### 2.6. The Pharmacological Pair Enhances the Suppression of Prostate Cancer 22Rv1 Xenograft Growth

The efficacy of the pharmacological pair was assessed in male nude mice bearing 22Rv1 tumors (Figure 3). The combination of C115H MGL-Dz and propiin led to a significantly higher reduction in tumor volume (139 ± 73 mm^3^; TGI = 70%; *p* = 0.043) compared with the vehicle group (460 ± 208 mm^3^). C115H MGL-Dz/PBS reduced tumor growth by 24% from the first day of treatment compared to the control (final tumor volume of 352 ± 245 mm^3^, *p* = 0.165) when the differences were not statistically significant. On the last day of therapy, tumor growth in the Group 4 was reduced by 5% compared to the vehicle group, and the effect was not significant (*p* > 0.05). Body weight was assessed to ensure that the treatment was safe. No significant changes in body mass were observed across all groups during the 10-day medication therapy, indicating that the dosage regimens were well tolerated and had no obvious adverse effects.

## 3. Discussion

Prodrugs are compounds that are designed to enhance the pharmacokinetic profile of a drug. The enzymes commonly utilized in EPT are non-mammalian or have low systemic abundance in the organism [13]. Since the 1980s, MGLs from different sources have been evaluated as anticancer agents [14,15]. The absence of MGL in mammals makes the enzyme a suitable candidate for EPT.

G protein-coupled estrogen receptor (GPER), also known as GPR30, was found to be a new membrane estrogen receptor that responds to estrogens in a nongenomic manner by stimulation of second messenger pathways [16]. Isoflavones, including daidzein and genistein, are known as GPR30 ligands [17].

Based on these findings, we prepared activated daidzein and conjugated it to C115H MGL to yield C115H MGL-Dz. Modification of C115H MGL by daidzein had practically no effect on the steady-state kinetic parameters of the conjugated enzyme. The catalytic efficiencies of conjugates in the β-elimination reaction of S-(allyl/alkyl)-L-cysteine sulfoxides were close to the parameters determined for the non-conjugated enzyme [11]. An increase in the affinity of the modified enzyme to substrates was observed, along with a slight decrease in the k_cat_ values. As a result, covalent attachment of three molecules of daidzein to the enzyme had no effect on the catalytic activity of C115H MGL.

The non-toxic nature of the most known S-allyl-L-cysteine sulfoxide, alliin, has been recognized by the FDA as a GRAS substance. This fact encouraged us to study other sulfoxides as prodrugs in EPT. Allicin is the most effectively studied thiosulfinate. The effects of allicin on various cancer cells and the cytotoxicity of allicin were shown on 3T3 (murine fibroblast cell lines), A549 (human lung carcinoma), HT29 (human colon carcinoma), MCF7 (human breast cancer), HUVEC (human umbilical vein endothelial cell) and human acute myeloid leukemia cell lines [18]. Previous research has discovered that allicin not only suppresses malignant cell growth, but also stimulates cell apoptosis in a variety of cancers, including glioblastoma, breast cancer, gastric carcinoma, pancreatic cancer cells (MIA-PaCA2) and hepatocellular carcinoma (HepG2) [19].

Synthetic thiosulfinates have previously been found to inhibit cell growth and proliferation of Caco-2, LS174T, and HCT116 (colon cancer cell lines) [20]. The antimetastatic properties of thiosulfinates have also been demonstrated [21]. Furthermore, allicin has been found to interact with the mitochondrial electron transport chain in cells and induce apoptosis, making it a promising target for cancer treatment [19].

Targeted delivery of the C115H MGL as a component of pharmacological pairs to generate thiosulfinates directly at the surface of tumor cells may be successful in reducing the toxic effects of thiosulfinates. Earlier, we demonstrated the absence of toxic effects of pharmacological pair C115H MGL/propiin where the combination was prepared for the targeted delivery prodrug (propiin) to selectively kill tumor cells in vivo [22]. We have explored the effect of dialkyl thiosulfinates generated by the C115H MGL/S-alkyl-L cysteine sulfoxide system against MCF7, SW620, SCOV-3 [11], SKBR3, MDA-MB-231 and T-47D [22] cell lines. In addition, C115H MGL-Dz/propiin was shown to cause a significant increase in anticancer activity against human breast cancer cell line SKBR3 in vivo [22].

In the present report, the antiproliferative capacity of dipropyl thiosulfate generated by the pharmacological pair C115H MGL-Dz/propiin was evaluated in vitro against cancer cells derived from solid tumors: COLO205, HCT-116, HT29 (colon cancer cells); Panc1, MIA-PaCa-2 (pancreatic cancer cells); and 22Rv1, DU-145 and PC3 (prostate cancer cells). All cell lines were found to be sensitive to dipropyl thiosulfinate and a significantly decreased IC_50_ value vs. the control group was demonstrated. IC_50_ values ranged from 18.6 to 66.3 µM. In vivo, the pharmacological pair effectively inhibited 22Rv1, Panc1, MIA-PaCa2 and HT29 tumor growth. We speculate that C115H MGL-Dz can bind with membrane receptors in the cells, producing high local concentration of cytotoxic thiosulfinates. In HT-29 cells, Dz can bind with GPER1 and/or ERβ [23]. Furthermore, the overexpression of GPER1 in Panc1 permits Dz to bind with ER- Panc1 more efficiently than ER+ MIA-PaCa2 in pancreatic cancer [24,25]. In prostate cancer, GPER1 has been demonstrated in Pc3 cells [26]. On the other hand, there has been a paucity of research on the existence of GPER1 expression in 22Rv1. Given the significant sensitivity of 22Rv1 cells to the C115H MGL-Dz treatment discovered in the current investigation, we surmise that GPER1 is overexpressed in 22Rv1. However, the treatment did not significantly decrease the tumor volume in comparison with the control group SW620 due to the low expression of daidzein-binding receptors on the cells’ surface. None of the mice exhibited any signs of physical discomfort during the treatment or follow-up periods. These results suggest that the pharmacological pair C115H MGL-Dz/propiin may significantly inhibit tumor growth both in vitro and in vivo.

## 4. Materials and Methods

### 4.1. Reagents

S-methyl-L-cysteine and nicotinamide adenine dinucleotide reduced form (NADH) were obtained from Sigma-Aldrich (Schnelldorf, Germany). PLP and D,L-dithiothreitol (DTT) were purchased from Serva (Heidelberg, Germany). Propiin (S-propyl-L-cysteine sulfoxide), methiin (S-methyl-L-cysteine sulfoxide), and alliin (S-allyl-L-cysteine sulfoxide) were prepared according to [27]. GE Healthcare (Styria, Austria) provided ethylenediaminetetraacetic acid (EDTA). Penicillin was purchased from PanEco (Moscow, Russia). 2-Nitro-5-thiobenzoate (NTB) was prepared according to [28]. The plasmid with the gene of *C. novyi* C115H MGL was obtained from Eurogene (Moscow, Russia). Sigma-Aldrich (Schnelldorf, Germany) provided 3-(4,5-dimethylthiazol-2-yl)-2,5-diphenyltetrazolium bromide (MTT). Trypan blue, phosphate-buffered saline (PBS) and dimethyl sulfoxide (DMSO) were purchased from PanEco (Moscow, Russia). Fetal bovine serum was obtained from HyClone (Logan, UT, USA) along with flasks and plates were purchased from Nunc (Moscow, Russia).

### 4.2. Preparation of C115H MGL and Enzymes Assays

The recombinant C115H MGL was obtained from *Escherichia coli* BL21 (DE3) cells containing pET28a plasmid with the inserted gene of the mutant form of *C. novyi* C115H MGL. The cells were grown and the enzyme was isolated as described in [29]. The assays were performed by measuring the rate of the β-elimination reaction in 100 mM potassium buffer solution, pH 8.0, containing 1 mM EDTA, 0.1 mM PLP, 1 mM DTT and 100 mM S-methyl-L-cysteine [29]. The rate of pyruvate formation was determined in the coupled reaction with LDH by reducing the absorption of NADH at 340 nm (ε = 6220 M^−1^ cm^−1^) at 37 °C. The amount of the enzyme that catalyzes the formation of 1.0 μmol/min of pyruvate was taken as one unit of activity [30].

### 4.3. Steady-State Kinetics

The kinetic parameters of β-elimination reaction of S-(alkyl/allyl)-substituted L-cysteine sulfoxides, S-methyl-L-cysteine catalyzed by the mutant form, and the conjugated enzyme were calculated as mentioned above using various amounts of the substrates. EnzFitter software was used to process the data using the Michaelis–Menten equation. A subunit’s molecular mass 43.25 kDa was used for the calculations.

### 4.4. Preparation of C115H MGL/Propiin Mixture and Quantitative Determination of Dipropyl Thiosulfinate

In total, 20 mg/mL of propiin was dissolved in 50 mM of potassium phosphate buffer, pH 7.0, containing 0.1 mM PLP, then combined with C115H MGL (0.14 U/mL) and incubated at 37 °C for 1 h. The NTB assay was used to quantify the concentration of thiosulfinate produced in the reaction mixture [28].

### 4.5. Synthesis of C115H MGL-Dz

The N-hydroxysuccinimide ester of 7-(O)-carboxymethyl (daidzein-NHS) was synthesized, as previously reported [31]. In total, 400 mg of C115H MGL was added to 400 µL activated daidzein in 50 mL of 50 mM potassium phosphate buffer, pH 7.0 containing 1 mM DTT, 0.1 mM PLP and 10% glycerol, and then stirred for 4 h at 4 °C. The C115H MGL-Dz was washed thoroughly three times with potassium buffer. The number of daidzein residues linked to C115H MGL was calculated using a technique devised for determining the bound steroid residues in macromolecular steroid conjugates [32] using the UV absorbance of daidzein at 265 nm (ε = 23,262 M^−1^cm^−1^) and the absorbance of C115H MGL at the same wavelength.

### 4.6. Cell Lines and Cytotoxicity Evaluation

Human colon adenocarcinoma cell lines (HT29, COLO205 and HCT116), human pancreatic cancer (Panc1 and MIA-PaCa2) and human prostate cancer cell lines (22Rv1, DU-145 and PC3) were purchased from ATCC (Manassas, VA, USA). Cancer cells were routinely grown in Gibco Dulbecco’s Modified Eagle Medium (DMEM) culture medium, supplemented with 10% fetal bovine serum, glutamine, and 100 U/mL penicillin. HT29, COLO205, HCT116, Panc1, MIA-PaCa2, 22Rv1, DU-145 and PC3 cells were grown in flasks in DMEM fresh culture medium with supplements at 37 °C and 5% CO_2_. Cells were grown as monolayer cultures, and the cells in the exponential growth phase were trypsinized and suspended in DMEM medium supplemented.

#### 4.6.1. The Cytotoxicity of the C115H MGL-Dz in the Presence of Propiin

In 96-well plates, cells were seeded at a density of (6–10) × 10^3^ cells per well and incubated for 24 h at 37 °C in humidified 5% CO_2_. C115H MGL-Dz was added in two-fold serial (1–128 μM) dilutions to pre-incubated cells at 37 °C for 30 min. Unbound conjugates were removed by washing the cells 3 times in DMEM culture medium before adding 1 mg/mL propiin and incubating for 24 h under optimum growth conditions at 37 °C in humidified 5% CO_2_.

#### 4.6.2. The Cytotoxicity of Dipropyl Thiosulfinate Generated by the Pharmacological Pair C115H MGL/Propiin

To evaluate the cytotoxicity of dipropyl thiosulfinate in vitro, we placed cells in (3–8) × 10^3^ cells/mL concentrations in 96-well culture plates for 24 h. They were then exposed to different concentrations (1–300 µg/mL) of dipropyl thiosulfinate obtained from the mixture of C115H MGL/sulfoxide for 72 h.

In both methods, cells were counted after treatment with Trypan blue solution (0.4%). In control wells with untreated cells, only PBS was added. Cell viability was measured by the standard MTT test [33]. The absorbance was measured at 540 nm using a Multiskan™ FC microplate photometer and Skanlt software 6.1 RE for microplate reader, both from Thermo Scientific (Waltham, MA, USA).

### 4.7. Animals and Evaluation of Antitumor Activity

Six- to ten-week old BALB/c nude mice were used for the study: 5 mice per group. Animals were housed in the N.N. Blokhin National Medical Research Center of Oncology antiviral temperature-controlled facility (25 ± 2 °C), fed a standard chow diet, and given unlimited access to food and water. All animal experiments followed EU directives on the protection of animals used for scientific purposes. Institutional guidelines for the proper and humane use of animals in research were followed, and the animal studies were approved by the N.N. Blokhin National Medical Research Center of Oncology local ethics committee for animal trials. Cancer cells 10^6^ were suspended in 0.3 mL DMEM and transplanted s.c. by a trocar needle. When tumor volume reached approximately ~100 mm^3^, mice were randomized into four experimental groups: Group 1 was injected with PBS only (control); Group 2 Conjugate C115H MGL-Dz was injected i.p. (30 U in 200 µL phosphate buffer (PBS) containing 20 µM PLP, pH 7.4) 1.5 h later followed by a propiin intratumoral injection (3 mg in 100 µL PBS); Group 3 was injected as Group 2, but propiin was substituted with PBS; and Group 4 was injected with unconjugated C115H MGL instead of C115H MGL-Dz conjugate followed by a propiin injection. Treatment was carried out for ten days at 24 h intervals. Tumor volume and body weight were measured 2 times per week to track the effect of the treatment. The tumor volumes were determined using the formula:(Length × width × heigth × π)/6

Ref. [34], the lag in tumor growth was used to evaluate treatment response. Growth curves were constructed using the average tumor volumes within each experimental group. The anticancer efficacy of C115H MGL-Dz with propiin in situ or other therapeutic groups was calculated using tumor growth inhibition (TGI %), using the formula:TGI% = ((Vc − Ve)/Vc) × 100%,
where Vc and Ve were substituted with the average tumor volume (mm^3^) in the control and experimental groups, respectively.

The tolerability of the therapy was determined by measuring body weight, monitoring appearance daily, and autopsy findings.

### 4.8. Statistical Analysis

In vitro experiments were carried out in triplicate. Graphpad prism version 9.0 was used to determine the IC_50_. The data of IC_50_ are presented as mean ± standard deviation (SD). Statistical analyses of in vivo studies were performed using SPSS statistics version 25.0 (IBM, New York City, NY, USA) using non-parametrical Mann–Whitney U-test. Both line graphs and bar graphs expressed mean values and error bars as standard deviation (SD). A probability value of *p* was calculated between the control group and each treatment group. *p* < 0.05 was considered statistically significant.

## 5. Conclusions

In this study, the conjugates of C115H MGL with daidzein were obtained and characterized. In vitro cytotoxicity of dipropyl thiosulfinate produced by the pharmacological pair C115H MGL-Dz/propiin was demonstrated on different cancer cells, and the conjugates of C115H MGL-Dz in combination with propiin were shown to suppress cancer cells growth in vitro and in vivo. 22Rv1, Panc1, MIA-PaCa2 and HT29 xenografts were sensitive to treatment with pharmacological pair C115H MGL-Dz/propiin. Our findings may inspire further research work into the role of pharmacological pairs as a new approach for cancer treatment.

## Data Availability

The data that support the findings of this study are available from the corresponding author upon reasonable request.

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
