# Peer review of "Methionine γ-Lyase-Daidzein in Combination with S-Propyl-L-cysteine Sulfoxide as a Targeted Prodrug Enzyme System for Malignant Solid Tumor Xenografts"

_ijms, 2022, doi:10.3390/ijms231912048_

Round 1

Reviewer 1 Report

In the manuscript entitled as “Methionine γ-lyase-daidzein in combination with 2 S-propyl-L-cysteine sulfoxide as a targeted prodrug enzyme 3 system for malignant solid tumor xenografts” by Louay Abo Qoura and coworkers investigate the administration of an enzyme-substrate combination to generate a cytotoxic compound which could exert cytotoxic effect toward cancer cells. I think the work is original, and the experiments (both enzymology and anti-cancer tests) are performed accurately. However, I have concerns regarding some of the experiments they performed, which issues I collected in the points below:

1. The Authors should explain better why they functionalised the enzyme with the daidzein moiety. I believe that it helps to stabilise the complex on the surface of the cancer cells, however it is not clearly written in the manuscript. Also, previous scientific works should be cited to better explain their choice of modification (e.g. cellular targets of daidzein).

2. After incubating the cells with C115H MGL-Dt for 30 min, the unbound enzyme was washed out. It would be important to approximate the binding efficiency of the applied enzyme on the different cell lines. Is is possible that the different toxicity profiles on the various cell lines/xenografts are related to the different binding affinity of the enzyme on the surface? Using western blot analysis of the supernatant vs cell lysates, the enzyme-binding affinity could be easily measured. 

3. Following binding, does the enzyme get internalised by the cells? How does it affect its activity? Is the prodrug cell membrane permeable?

4. After washing out the non-bound enzyme, the Authors added the pro-drug propiin to the cells, however it is not mentioned in the manuscript how long the cells were then incubated with the prodrug (and subsequently, with the converted active compound). This should be corrected.

5. The IC50 values of the prodrug itself without the modifying enzyme should be measured on the applied cell lines. 

6. It would be crucial to include statistical tests for the in vivo xenograft test where the Authors compare Group 2 with Group 3. In the present manuscript, the Authors mostly compare the different experimental groups with the controls. However, by comparing Group 2 and 3, the effect of the prodrug->active compound transformation could be tested. I think it would be essential, to prove that truly the drug activation triggers the observed anti-cancer effects. 

Author Response

We appreciate the time and effort that you dedicated to providing feedback on our manuscript and are grateful for the insightful comments on and valuable improvements to our paper. We have incorporated most of the suggestions in the manuscript. Those changes are highlighted within the manuscript. Please see below, in blue, for a point-by-point response to your comments. All page numbers refer to the revised manuscript file with tracked changes.

Reviewer 2 Report

Abo Qoura et al. sumitted an original paper describing the antitumor effects of dipropyl thiosulfinate derived from methioninie y-lyase-daidzein or methioninie y-lyase in combination with S-propyl-L-cysteine sulfoxide (propiin) in in vitro and in vivo models. 

  1. The authors claim that the above combination is a source of thiosulfinates generation near cancer cells, which limits their side effects when used as synthetic compounds. The authors should use such control in all their experiments to verify whether the observed effects are specifically derived from dipropyl thiosulfinate. 
  2. The authors did not provide any evidence of the dipropyl thiosulfinate mechanism of action. As far as I know, the primary mechanism is based on ROS generation. Please, use the antioxidant to verify whether the activity of dipropyl thiosulfinate will be reduced (in vitro studies). 
  3. Further, the authors did not explain the significant differences in various cancer cells' sensitivity to applied treatment. According to Table 2, IC50 for colon cancer cells was not detected. Is that mean that these cell lines were resistant to combination treatment? Please provide a possible explanation. 
  4. According to Fig 4, the mice body weight was significantly reduced in all experimental groups compared to the control, suggesting toxicity. Please provide the toxicology data.
  5. The IC50 values of C115H MGL/PBS range from 249 to > 6800. Please explain such significant differences. Are there any non-specific mechanisms of action?
  6. Significant differences between treatment and control points should be marked in Figures (as * or ** or ***).
  7. The authors use the data from reference 6. Did the authors obtain permission from the Editorial Office to publish these data once again?

Author Response

(The authors gave the same response as above.)

Round 2

Reviewer 2 Report

The Authors addressed all suggestions and updated the manuscript or provided an explanation. I recommend the acceptance of the manuscript.